# Effectiveness of Pulmonary Rehabilitation in Severe and Critically Ill COVID-19 Patients: A Controlled Study

**DOI:** 10.3390/ijerph18178956

**Published:** 2021-08-25

**Authors:** Gilbert Büsching, Zhongxing Zhang, Jean-Paul Schmid, Thomas Sigrist, Ramin Khatami

**Affiliations:** 1Barmelweid Academy, Klinik Barmelweid AG, 5017 Barmelweid, Switzerland; ramin.khatami@barmelweid.ch; 2Department of Therapeutics, Klinik Barmelweid AG, 5017 Barmelweid, Switzerland; 3Center for Sleep Medicine, Sleep Research and Epileptology, Klinik Barmelweid AG, 5017 Barmelweid, Switzerland; 4Department of Cardiology, Klinik Barmelweid AG, 5017 Barmelweid, Switzerland; jean-paul.schmid@kliniken-valens.ch; 5Department of Pulmonology, Klinik Barmelweid AG, 5017 Barmelweid, Switzerland; thomas.sigrist@barmelweid.ch; 6Department of Neurology, Inselspital, Bern University Hospital, University of Bern, 3012 Bern, Switzerland

**Keywords:** COVID-19, pulmonary rehabilitation, pneumonia, 6-MWT, CRQ, FIM

## Abstract

Background: Severe and critically ill COVID-19 patients frequently need pulmonary rehabilitation (PR) after hospitalization. However, little is known about the effectiveness of PR in COVID-19 patients. Methods: We compared the performances in the six-min walk test (6MWT), chronic respiratory questionnaire (CRQ), and Functional Independence Measure (FIM) from inpatient PR between 51 COVID-19 patients and 51 other patients with common pneumonia. We used multivariate linear regression controlled for baseline values at entrance, age, sex, and cumulative illness rating scale. The odds ratios (ORs) of non-improvement/improvement in 6MWT (>30-m) and CRQ (>10-point) at discharge were compared between the two groups (Fisher’s exact test). Results: The two groups had similar improvements in 6MWT and CRQ, but the COVID-19 group achieved a 4-point higher FIM (*p*-value = 0.004). The OR of non-improvement/improvement in 6MWT was 0.30 (*p*-value = 0.13) between COVID-19 and controls; however, the odds of non-improvement in CRQ tended to be 3.02 times higher (*p*-value = 0.075) in COVID-19 patients. Severe and critical COVID-19 patients had similar rehabilitation outcomes. Conclusions: Inpatient PR can effectively improve physical functions and life quality in COVID-19 patients, irrespective of disease severity. Whether the relatively low gains in CRQ is an indicator of chronic disease development in COVID-19 patients needs further studies.

## 1. Introduction

Severe SARS-CoV-2 infection often leads to hospitalization due to COVID-19 pneumonia. A considerable proportion of these patients need mechanical ventilation at the intensive care unit (ICU) due to acute respiratory distress and will develop ICU-acquired weakness [1,2]. Recently, increasing evidence suggests that, after hospitalization, COVID-19 patients may still have abnormal pulmonary function [3], low physical functioning and impaired performances of daily life activities [4], persistent fatigue and sleep difficulties [5,6]. Therefore there are increasing rehabilitation needs for COVID-19 patients who continue to suffer from the long-term consequences of COVID-19. A European Respiratory Society- and American Thoracic Society-coordinated international task force published interim guidance on rehabilitation for COVID-19 patients during hospitalization and the post-hospital period [7]. However, this guidance is not based on data but reflect experts’ recommendations due to the lack of data on the safety and efficacy of rehabilitation programs. Since the number of patients with SARS-CoV-2 infection is exponentially increasing, COVID-specific PR programs are warranted based on a broad data basis including clinical courses of different virus variants.

Hermann et al. first reported that current pulmonary rehabilitation (PR) developed for pneumonia patients can improve physical performance in the six-min walk test (6MWT) of critical COVID-19 patients taken from a Swiss cohort of 28 patients who have been admitted to the ICU between March and May 2020 [8]. Subsequent studies on post COVID-19 patients reported on more variable PR outcomes probably because many of these studies focused on patients’ characteristics after acute illness [3,9], used different outcome assessments (e.g., 6MWT vs. sit-to-stand test, Functional Independence Measure (FIM) vs. Barthel Index) [10,11,12,13,14], different PR settings (e.g., immediately post-ICU [12] vs. discharged from acute hospitalization [10,14]), and had a lack of control groups. In addition, the assessment of the minimal clinically important difference (MCID) is missing in most of previous studies, making it difficult to estimate the relevance of clinical improvement [15,16]. Thus, it remains unclear whether rehabilitation programs applied to COVID-19 patients can achieve similar improvements in the physical functioning and quality of life compared to other pneumonia patients.

In addition, the mutation of the virus is currently probably challenging the development of specific rehabilitation programs [17,18,19,20,21,22,23,24]. Increasing evidence suggests that the spread and toxicity of the variants are getting stronger [21,25,26], and increasing the complexity of cases over time (e.g., longer ICU stay of patients before the inpatient rehabilitations). Although an increasing amount of data of PR in post COVID-19 is now available, most of the data contain mixtures of different virus variants. Unfortunately, we only have very limited data from the beginning of this pandemic, which corresponds to the original virus, probably because a shortage of medical resources at the beginning of this pandemic in the spring of 2020 prevented researchers from timely data collection.

In this study, we therefore retrospectively investigated the clinical course of COVID-19 patients during PR and compared the outcomes of physical performance (i.e., 6MWT), FIM [27], and the chronic respiratory questionnaire (CRQ) [28,29] to matched cases suffering from other types of pneumonia who attended the same rehabilitation program. Our patients were mostly to be infected by the original virus because they were infected during the first wave of the pandemic in the spring of 2020. Special interest was given to the proportion of patients who failed to achieve a clinically relevant improvement as indexed by the MCID and the impact of ICU-treatment on their outcome.

## 2. Materials and Methods

We retrospectively investigated a cohort of 51 COVID-19 patients with pneumonia referred from acute care hospitals to inpatient pulmonary rehabilitation at the Department of Pulmonology, Clinic Barmelweid AG between 23rd March and 29th May 2020. The patients started the inpatient PR program immediately after discharge (confirmed by two negative PCR-test within 48 h) from acute care hospitals. We compared these COVID-19 patients with 51 consecutive patients with common pneumonia (48 patients with community-acquired-pneumonia, 2 patients with aspiration and 1 patient with infarct pneumonia), who did the same PR protocol in 2019 in our clinic. Individuals in the common pneumonia control group were selected according to the following criteria: (1) age greater than 40 years; (2) no history of thoracic and/ or pulmonary surgery; (3) no repetition of the PR program in 2019. According to Swiss national ethical and legal regulation, ethical approval by the local Ethics Committee was not needed for this study. All patients gave general consent to use their data for research purposes and all data complied with relevant data protection and privacy regulations.

Patients received standard PR as defined by the Swiss Respiratory Society [30,31]. It includes entry and discharge assessment, goal definition, therapy planning, a minimum of 540 minutes of patient education and therapy in single and group settings. Physiotherapeutic units consist of cardiopulmonary training (e.g., cycling, guided walking) and strength exercise (e.g., weight training on machines, free weight, elastic resistance bands) with optional oxygen supply if needed, breathing exercises (e.g., deep breathing, sputum evacuation), relaxation techniques (e.g., progressive muscle relaxation), and, if indicated, psychological counseling, speech, nutritional, and occupational therapy and social services. Physical performance (6MWT), CRQ, and FIM at entrance and at discharge from the PR program as well as improvements between entrance and discharge, were compared between the two groups. The numbers of patients whose improvements reached the established MCID: 30-m for 6 MWT [32] and 10 points for CRQ were also calculated in the two groups. We chose the 10-point upper threshold for CRQ because we used the original CRQ with a total score of 140 points. A 10-point improvement corresponds to a 0.5-point MCID when converted to the 1–7 point system [33].

### Data Analysis

Odds ratio (OR) and its 95% confidence interval (CI) were calculated to compare binary parameters between the control group and the COVID-19 group (Fisher’s exact test). The non-parametric Wilcoxon rank-sum test was used to compare the numerical variables between the two groups if their distributions were not normal (tested by the Shapiro–Wilk test). For numerical variables with normal distributions, we first compared the variances between the two groups (F-test), followed by a two-sample *t*-test or a Welch’s two- sample *t*-test depending on the equal or unequal variance between the two groups. Fisher’s exact test was applied to test differences in the OR of non-improvement vs. improvement (i.e., failed vs. achieved MCID) in 6MWT and CRQ at discharge between the two groups. The significance level was *p*-value < 0.05.

Linear regressions were used to compare whether the 6MWT distance, FIM, and CRQ scores remained different between the two groups at discharge, after controlling their values at entrance, age, sex, and cumulative illness rating scale (CIRS). We further tested whether the outcomes of our PR program were different between severe and critical patients (i.e., non-ICU vs. ICU) using linear regression after controlling the aforementioned variables. This subgroup analysis can test whether a similar efficacy of the rehabilitation outcome can be also achieved in critical COVID-19 patients admitted to ICU during hospitalization. All statistical analyses were done using R (version 3.2.4).

## 3. Results

### 3.1. Patient Demography

Patients’ demographics at entrance are shown in Table 1. Most of our COVID-19 patients were male and younger than controls. They had higher CIRS scores at entrance, probably because more COVID-19 patients were treated at ICU and had mechanical ventilation compared to the control group. COVID-19 patients had higher odds to have the comorbidities of hypertension, diabetes, and acute respiratory distress syndrome (ARDS), while the control group had higher odds of having a comorbid chronic obstructive pulmonary disease (COPD).

### 3.2. Outcome of Rehabilitation: COVID-19 Patients vs. Common Pneumonia Patients

At baseline, the two groups had similar performance in 6MWT, while COVID-19 patients had better CRQ and FIM scores (Table 2).

The changes in the performances of 6MWT, CRQ, and FIM compared to baseline in each patient are shown in Figure 1, indicating that the majority of patients improved in both groups. Paired *t*-tests confirmed that both groups had significant improvements in 6MWT, CRQ, and FIM (Table 2). At discharge, COVID-19 patients achieved better performances in 6MWT and FIM but revealed similar CRQ scores after our PR program compared to the control group (Table 2). Likewise, COVID-19 patients tended to have a higher degree of improvement in 6MWT compared to the control group, whereas the increase in CRQ and FIM were similar in both groups with no significant differences (Table 2).

We checked the fatigue score of CRQ in our COVID-19 patients. Twenty-five patients performed the CRQ at entrance, and their mean and median CRQ-fatigue scores were both 16 points, with IQR between 12 and 21 points. At discharge, 36 patients performed the CRQ, and their mean and median CRQ-fatigue scores were 20 and 19 points, respectively, with IQR between 17 and 22 points. A paired *t*-test suggested that the CRQ-fatigue score was significantly improved by 4.2 points (*n* = 21, *p*-value < 0.0001) after PR.

Considering the differences in age, sex, and CIRS scales at baseline between the two groups, we performed a regression analysis to test whether the outcomes at discharge were still significantly different after controlling for these variables and for the baseline performances. The results are shown in Table 3. The data of 37 COVID-19 and 47 control patients were used to build the regression model of 6MWT after deleting patients with missing values and outliers, defined by a nonparametric boxplot. The numbers of patients were 21 and 51 for the model of CRQ and 44 and 49 for the model of FIM, respectively. There were no significant differences in the 6MWT performance and CRQ at discharge between these two groups, but COVID-19 patients had on average 4.16 points more (*p*-value = 0.00364) on their FIM score compared to the control group.

Next, we compared the number of patients who did not achieve a clinically relevant outcome according to the MCID. A total of 7.5% (3/40) COVID-19 and 21.3% (10/47) of the control patients failed to meet a MCID of 6MWT after the PR program. A Fisher’s exact test showed that the OR of non-improvement/improvement between the COVID-19 and control groups was 0.30 (95% CI: 0.05–1.31, *p*-value = 0.13). So the odds of no improvement in 6MWT in the COVID-19 group were not different from that of the control group, although the proportion appeared to be smaller in the COVID-19 group (i.e., 7.5% vs. 21.3%).

An even higher proportion, 42.9% (9/21), of COVID-19 patients did not reach an MCID of CRQ after the inpatient PR. This number was 19.6% (10/51) in the control group, resulting in an OR of 3.02 (95% CI: 0.87-10.64, *p*-value = 0.075) of non-improvement/improvement in CRQ between the COVID-19 and control groups. Thus, the odds of no effect in improving CRQ showed a trend 3.02 times higher in the COVID-19 group compared to the control group. Of the nine COVID-19 patients who failed in MCID, six were critically ill at ICU and intubated for mechanical ventilation; six patients were male, and four patients were without ARDS. These smaller numbers of patients prevented us from further exploring which factors predict failure to improve CRQ via regression analysis.

### 3.3. Outcome of Rehabilitation: COVID-19 Patients ICU vs. Non-ICU

The data of 23 critical COVID-19 patients who were treated at ICU and 14 severe COVID-19 patients who were not at ICU were used to build a regression model of 6MWT, after deleting patients with missing values and outliers. The dependent variable was the 6MWT at discharge, and age, sex and CIRC scales and baseline 6MWT performance were controlled in the model. The results are shown in Table 4 below. There was no significant difference in the 6MWT at discharge between the two subgroups, indicating that the physical functioning outcome of our rehabilitation program is similar between critical and severe COVID-19 patients.

Similarly, 13 and eight COVID-19 patients treated at ICU vs. not treated at ICU were used to predict the CRQ at discharge (Table 4). Although the fitting of the model was relatively poor (i.e., *R*^2^ = 0.40), probably due to the relatively small number of patients, the results still suggested that there was no difference in CRQ at discharge between the two subgroups.

The numbers of patients were 26 and 18 in the ICU subgroup and non-ICU subgroup, respectively, for the model of FIM at discharge. Again, there was no significant difference in FIM scores at discharge between these two subgroups (Table 4).

## 4. Discussion

In this comparative study, we show that survivors of COVID-19 pneumonia infected by the original virus benefit from a standardized multidisciplinary inpatient rehabilitation program [30,34] in terms of improved physical capacity, disease-related quality of life, fatigue, and functional outcome. When compared to patients with common pneumonia who underwent the same rehabilitation program, COVID-19 patients achieve better outcomes in physical capacity and gain a similar disease-related disability and quality of life, even though the COVID-19 group has a higher cumulative illness rating scale and a larger proportion of patients who had been admitted at ICU and needed mechanical ventilation. Regression analyses show that successful rehabilitation outcomes are still apparent after controlling for confounders such as age, sex, cumulative illness rating scale, and different baseline values at the beginning of the rehabilitation program. We conclude that inpatient rehabilitation is effective and suitable for most COVID-19 patients admitted from hospitals for acute care [4]. Remarkably, critically ill COVID-19 patients treated at ICU obtained a similar outcome after rehabilitation compared to COVID-19 patients not mechanically ventilated at ICU. Thus pulmonary rehabilitation allows for a fast improvement even after critically ill COVID-19 infection.

The strength of our study is twofold. We compared the effectiveness of rehabilitation in COVID-19 patients most likely infected by the original virus to that of one of the most common types of pneumonia patients and considered clinically relevant improvement after PR an outcome parameter. Based on our data, we are impressed by two findings: First, there is a discrepancy between the high physical improvement of COVID-19 patients yet relatively low gains in disease-related quality of life (CRQ) compared to the common pneumonia group. This finding is unlikely to be explained by a ceiling effect of the CRQ since COVID-19 patients started at a high level of CRQ—as they did for physical 6MWT performance—but stayed far below the maximum scores at the end of the PR program. Second, compared to controls, we found a relatively high proportion, 42.9%, of COVID-19 patients who failed to improve CRQ during rehabilitation. The reasons why PR was not effective in such a considerable proportion of patients cannot be fully explained by our data. These patients may have developed a chronic disease, or, alternatively, the PR program may need to be optimized for disease-specific aspects of COVID-19 rehabilitation. Recent available data indicate that COVID-19 patients after acute care hospitalization still suffer low physical functioning and impaired performances of daily life activities [4,6]. Whether this is related to persistent abnormal pulmonary function [3] and/or persistent fatigue [5,6] remains unclear. It is also well known that patients with ARDS (which was highly prevalent in our COVID-19 group) still show disability after years [2]. If a similar chronic course is confirmed for the COVID-19 pandemic, the whole world will need to consider a strong impact on health care systems in the future.

An interesting finding is that the PR outcome of our COVID-19 patients was not determined by the need for ICU treatment (Table 4). However, we hesitate to overinterpret this encouraging observation since the study design is neither appropriate nor sufficiently powered to draw direct conclusions due to the small numbers of patients and missing values (especially a large number of missing values in CRQ). Future controlled studies are warranted to show if ICU treatment is an important predictor of PR outcome.

Fatigue is the most frequent post-COVID-19 infection symptom reported in a recent study [6]. Our results suggest that an inpatient PR program immediately following acute hospitalization can significantly reduce the fatigue of COVID-19 patients. Unfortunately, the cut-off value defining the MCID of the CRQ fatigue category is unknown. We thus cannot evaluate the effectiveness of reducing fatigue using our inpatient PR program. Future studies using standard fatigue questionnaires in a large post-COVID-19 infection cohort are needed to further investigate the effectiveness of PR in relieving the fatigue symptom.

Comparing our study with the other studies, we suggest that more data are still needed in order to thoroughly assess the effectiveness of PR in patients post COVID-19 infection because of many factors, such as different PR settings, country differences, or different variants, that could be potential confounders. For example, in a recent study Gloeckl et al. showed a mean stay of 28 days at ICU of their patients recruited between November 2020 and January 2021 [10], while Hermann et al. and Wiertz et al. reported a shorter mean stay of 12–17 days at ICU in their patients [8] between March and May 2020 [9]. These studies done at different periods of this pandemic reported different lengths of stay at ICU [8,9,10], which could potentially influence the PR outcomes in their patients. Data from Switzerland [8,14] reported an improvement mainly between 100-m and 200-m in 6MWT. However, in France [12] some post-ICU patients could even show an improvement up to 400–500 m after inpatient PR. Our data fit the results published from Switzerland, probably because we had the same PR setting, in which patients started PR in rehabilitation centers after they were discharged from acute hospitals; whereas in the study from France their post-ICU patients started the baseline 6MWT after being immediately discharged from ICU. Therefore, the diverse results among different studies suggest that more data covering multiple aspects of PR in post COVID-19 patients are still urgently needed.

Our study may be limited by a selection bias, since we do not know the clinical course of patients referred to other institutions (e.g., nursery homes) or who went to ambulant rehabilitation. Another limitation is that the CRQ has not been validated for COVID-19 patients, which may also partly explain our results of relatively low MCID in CRQ in the COVID-19 group compared to the control group. But it is still likely to give an overview of the burden of disease [29] and should be urgently validated in COVID-19 patients considering the exponential growth of the pandemic. Finally, we have a lack of functional parameters from our patients, such as pulmonary function, blood gas analysis, and diffusing capacity, because we collected our patients’ data between March and May 2020 retrospectively. Future studies providing the aetiology of pneumonia and lung functional data are needed to better explore the PR efficiency in COVID-19 patients. However, our study still provides valuable evidence that PR is feasible and effective after COVID-19 hospital stay and ICU stay and compares favorably to PR in patients after a hospital stay for pneumonia. Our study will be helpful for future meta-analysis of the effectiveness of PR in COVID-19 patients by adding valuable data on the original virus infection, which may serve as ‘baseline’ effectiveness for recent studies and the subsequent studies of variants.

## 5. Conclusions

We conclude that inpatient rehabilitation is effective and suitable for most COVID-19 patients admitted from hospitals for acute care [4]. Remarkably, critically ill COVID-19 patients treated at ICU could obtain a similar outcome after rehabilitation, compared to COVID-19 patients not mechanically ventilated at ICU. Thus pulmonary rehabilitation allows for a fast improvement even after critically ill COVID-19 infection.

## 6. Highlights

The highlights of this study include:Our study demonstrates the effectiveness of inpatient PR in COVID-19 patients by both comparing the PR outcomes between the COVID-19 group and a control group with common pneumonia and assessing the minimal clinically important difference.Our study also shows similar effectiveness of inpatient PR between severe and critically ill COVID-19 patients.Our results suggest that the inpatient PR program immediately following acute hospitalization can significantly reduce the fatigue of COVID-19 patients.Our results suggest that COVID-19 patients discharged from acute care should attend the inpatient PR program in order to improve their physical function and quality of life, including fatigue.

## Figures and Tables

**Figure 1 ijerph-18-08956-f001:**
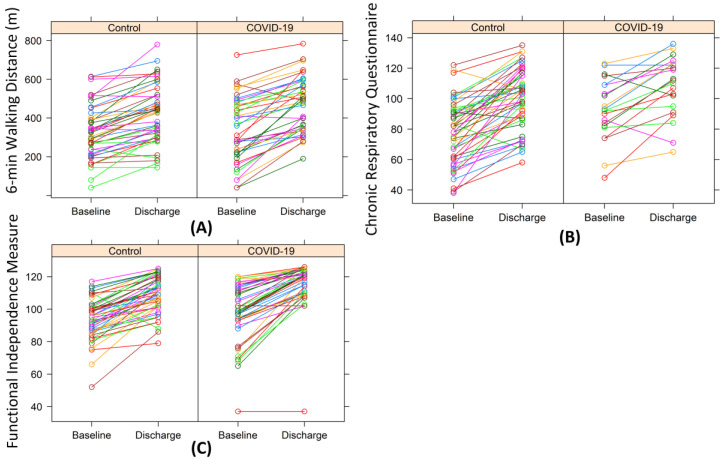
Change in performance of 6MWT, CRQ, and FIM. The performances of the 6-min walking test (**A**), chronic respiratory questionnaire (**B**), and Functional Independence Measure (**C**) in each patient at the baseline (i.e., entrance) and at the discharge from our pulmonary rehabilitation. In general, performance improved in both the control group (left) and the COVID-19 group (right).

**Table 1 ijerph-18-08956-t001:** Patient demography.

	Patients with COVID-19 (*n* = 51)	Patients with Other Pneumonia (*n* = 51)	Statistical Analysis
Male	38 (75%)	23 (45%)	OR: 3.5 [1.4, 9.0], *p* = 0.004
Age	65.8 ± 11.7, [59.0, 73.5]	69.8 ± 9.6, [65.0, 76.0]	*p* = 0.028
BMI	27.3 ± 4.9, [23.8, 30.1]	26.1 ± 6.5, [21.6, 29.3]	*p* = 0.28
Rehabilitation days	21.7 ± 5.8, [18.0, 27.0]	20.4 ± 5.4, [18.0, 21.5]	*p* = 0.20
Patients at ICU	30 (59%)	7 (14%)	OR: 8.8 [3.1, 27.7], *p* < 0.001
Patients intubated	27 (53%)	6 (12%)	OR: 8.2 [2.8, 27.9], *p* < 0.001
Intubation days	13.2 ± 7.1, [8.3, 15.0]	9.8 ± 8.3, [5.0, 12.0]	*p* = 0.15
CIRC	17.7 ± 11.3, [13, 20]	13.5 ± 5.9, [9, 18]	*p* = 0.026
Art. hypertension	30 (59%)	19 (37%)	OR: 2.4 [1.0, 5.8], *p* = 0.047
ARDS	26 (51%)	3 (6%)	OR: 16.2 [4.3, 91.5], *p* < 0.001
COPD	2 (4%)	25 (49%)	OR: 0.05 [0.005, 0.2], *p* < 0.001
Heart diseases	8 (16%)	10 (20%)	OR: 0.8 [0.2, 2.4], *p* = 0.8

Data are expressed as mean ± SD and IQR. SD: standard deviation IQR: interquartile range, BMI: body mass index, ICU: intensive care unit; CIRC: cumulative illness rating scale; COPD: chronic obstructive pulmonary disease; ARDS: acute respiratory distress syndrome; OR: odds ratio. Data are expressed as OR [95% confidence interval].

**Table 2 ijerph-18-08956-t002:** The comparisons of 6MWT, CRQ, and FIM between the two groups.

	Patients with COVID-19	Patients with Other Pneumonia	Statistical Analysis
6 MWT entrance	336.2 ± 169.3,	319.8 ± 135.5,	*p* = 0.61
	[222, 470], *n* = 41	[231, 389], *n* = 48	
6 MWT discharge	484.4 ± 146.6,	416.8 ± 144.8,	*p* = 0.026
	[346, 594], *n* = 45	[316, 503], *n* = 50	
6 MWT improvement	132.8 ± 92.9 *,	102 ± 73.3 *,	*p* = 0.088
	[72, 173], *n* = 40	[54, 138], *n* = 47	
CRQ entrance	91.7 ± 19.8,	77.9 ± 20.3,	*p* = 0.0063
	[82, 103], *n* = 25	[62, 91], *n* = 51	
CRQ discharge	105.8 ± 18.0,	100.2 ± 19.6,	*p* = 0.18
	[92.5, 120.5], *n* = 36	[88, 115], *n* = 51	
CRQ improvement	15.5 ± 15.2 *,	22.3 ± 16.9 *,	*p* = 0.12
	[5, 28], *n* = 21	[13.5, 32], *n* = 51	
FIM entrance	97.3 ± 17.4,	93.3 ± 12.3,	*p* = 0.035
	[93, 111], *n* = 4	[86, 100.5], *n* = 51	
FIM discharge	115.8 ± 14.0,	108.9 ± 10.9,	*p* < 0.001
	[111, 124], *n* = 45	[102, 117.5], *n* = 51	
FIM improvement	18.0 ± 11.4 *,	15.6 ± 9.6 *,	*p* = 0.48
	[10, 23], *n* = 45	[10, 21], *n* = 51	

Data are expressed as mean ± SD and IQR. 6MWT: six-min walk test; CRQ: chronic respiratory questionnaire, FIM: Functional Independence Measure, SD: standard deviation, IQR: interquartile range. * indicates the within-group improvement is significantly larger than 0, *p*-value < 0.0001.

**Table 3 ijerph-18-08956-t003:** The results of regression analysis of COVID-19 group vs. control group.

	6MWT at Discharge (Adjusted *R*^2^ = 0.75)			CRQ at Discharge (Adjusted *R*^2^ = 0.75)			FIM at Discharge (Adjusted *R*^2^ = 0.75)		
	Estimate	Std. Error	*p*-Value	Estimate	Std. Error	*p*-Value	Estimate	Std. Error	*p*-Value
Age	−2.06	0.81	0.0134	−0.23	0.18	0.202	−0.05	0.06	0.432
Sex:m	34.11	17.56	0.0557	1.33	3.77	0.726	2.45	1.35	0.072
6MWT entrance	0.70	0.06	<0.0001	/	/	/	/	/	/
CRQ entrance	/	/	/	0.62	0.09	<0.0001	/	/	/
FIM entrance	/	/	/	/	/	/	0.45	0.05	<0.0001
CIRC	−1.64	0.92	0.0787	−0.42	0.62	0.184	0.005	0.075	0.942
COVID-19: Control	26.55	17.61	0.136	−2.70	4.45	0.545	4.16	1.39	0.00364

6MWT: six-min walk test; CRQ: chronic respiratory questionnaire, FIM: Functional Independence Measure; CIRC: cumulative illness rating scale.

**Table 4 ijerph-18-08956-t004:** The results of a regression analysis of the COVID-19 ICU subgroup vs. the non-ICU subgroup.

	6MWT at Discharge (Adjusted *R*^2^ = 0.77)			CRQ at Discharge (Adjusted *R*^2^ = 0.40)			FIM at Discharge (Adjusted *R*^2^ = 0.61)		
	Estimate	Std. Error	*p*-Value	Estimate	Std. Error	*p*-Value	Estimate	Std. Error	*p*-Value
Age	−2.37	1.09	0.0379	−0.11	0.31	0.722	−0.11	0.069	0.11
Sex: m	55.38	26.77	0.047	8.91	7.96	0.280	3.71	1.62	0.027
6MWT entrance	0.57	0.073	<0.0001	/	/	/	/	/	/
CRQ entrance	/	/	/	0.68	0.18	0.0021	/	/	/
FIM entrance	/	/	/	/	/	/	0.36	0.051	<0.0001
CIRS	−1.50	0.98	0.14	−0.039	0.66	0.954	0.003	0.065	0.97
ICU: non-ICU	31.12	25.90	0.239	−5.09	7.87	0.528	1.89	1.63	0.25

6MWT: six-min walk test; CRQ: chronic respiratory questionnaire, FIM: Functional Independence Measure; CIRC: cumulative illness rating scale.

## Data Availability

Data supporting the reported results can be provided upon reasonable request by the corresponding author.

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
