# Peer review of "Effectiveness of Pulmonary Rehabilitation in Severe and Critically Ill COVID-19 Patients: A Controlled Study"

_ijerph, 2021, doi:10.3390/ijerph18178956_

Round 1

Reviewer 1 Report

Dear Authors.

I consider that the work presented by Büshing and colls, has been very well presented, a correct and fluent introduction, where they highlight the most important in pulmonary rehabilitation. However, I suggest clarifying the association of pulmonary rehabilitation with the different variants of the Sars-COV-2 virus, because it gives the impression that this is exactly what they are looking for, and finally it is not studied in this work.

The methodologies are clear and adequate for reproducibility. The statistics used are adequate and express in the text the informed consent of the patients; I did not observe an approval of any bioethics committee. The results are very well described, and the tables and figures shown are good.   

The discussion is well done, and contributes a lot to the interpretation of the results, it faces the limitations of the research and the results in a direct and straightforward way, but contributing to the sense of this research work.

A very good research work, thank you very much for publishing it.

Author Response

Dear reviewer,

thank you for appreciating our work. We provide a point-by-point response to your comments as shown below.

1.  I suggest clarifying the association of pulmonary rehabilitation with the different variants of the Sars-COV-2 virus, because it gives the impression that this is exactly what they are looking for, and finally it is not studied in this work.

Response: Thank you for your suggestion. We now have toned down the association of pulmonary rehabilitation with the different variants of the Sars-COV-2 virus in the discussion. The third paragraph of our Introduction which mentions the variants is now written as :

'In addition, the mutation of the virus is currently probably challenging the development of specific rehabilitation programs [18-25]. Increasing evidence suggest that the spread and toxicity of the variants are getting stronger [22, 26, 27], and increase complexity of cases over time (e.g., longer ICU stay of patients before the inpatient rehabilitations) at the entrance of PR in a rehabilitative setting. Although increasing number of data of PR in post COVID-19 is now available, most of the data contain mixtures of different virus variants. Unfortunately, we only have very limited data from the beginning of this pandemic which correspond to the original virus, probably because of shortage of medical resources at the beginning of this pandemic in 2020 spring prevented researchers from timely data collection.'

2.  I did not observe an approval of any bioethics committee. 

Response: Thank you for pointing out this issue.  According to Swiss national ethical and legal regulation, ethical approval by the local Ethics Committee was not needed for this study, because the database we analyzed is anonymised data that have already existed and all patients have given general consent to use their data for research purposes. Please see the reference with this link from the Swiss regulatory:

https://www.kofam.ch/en/applications-and-procedure/projects-that-do-not-require-authorisation/

We now add the following statement in the Method section:

'According to Swiss national ethical and legal regulation, ethical approval by the local Ethics Committee was not needed for this study. All patients gave general consent to use their data for research purposes and all data complied with relevant data protection and privacy regulations.'

Please see our changes in the attached manuscript.

Best regards

Gilbert Büsching and Zhongxing Zhang

Reviewer 2 Report

The manuscript titled: ¯ Effectiveness of pulmonary rehabilitation in severe and critically-ill COVID-19 patients: a controlled study.¯ is authored by Bushing G. et al.

The authors observed that inpatient rehabilitation is effective and suitable for most COVID-19 patients admitted from hospitals for acute care.

The subject is of great interest and the study is well conducted.

Some minor revisions are required to improve the message.

Although the statistical tests are well described, the authors are invited to add a subtitle (i.e. : DATA ANALYSIS or STATISTICAL ANALYSIS) to the material and methods section under which they can put the paragraph about statistics.

Please add a highlight section after the conclusion, with up to 5 bullet points to summarize the main discovery of the paper and/or to underly authors’ recommendations about patient care regarding the main conclusions of the paper.

Author Response

Dear reviewer,

thank you for appreciating our work. Here we provide a point-by-point response to your comments. 

1. Although the statistical tests are well described, the authors are invited to add a subtitle (i.e. : DATA ANALYSIS or STATISTICAL ANALYSIS) to the material and methods section under which they can put the paragraph about statistics.

Response:  Thank you for your suggestion. We added a subtitle 2.1. Data Analysis before the third paragraph of Materials and Methods. 

2. Please add a highlight section after the conclusion, with up to 5 bullet points to summarize the main discovery of the paper and/or to underly authors’ recommendations about patient care regarding the main conclusions of the paper.

Response: Thank you for your helpful suggestion. We now added the Highlight section after the conclusion, which is shown below:

' The highlights of this study include:

  1. Our study demonstrates the effectiveness of inpatient PR in COVID-19 patients by both comparing the PR outcomes between the COVID-19 group and a control group with common pneumonia, and assessing the minimal clinically important difference.
  2. Our study also shows similar effectiveness of inpatient PR between severe and critically ill COVID-19 patients.
  3. Our results suggest that the inpatient PR program immediately following the acute hospitalization can significantly reduce the fatigue of COVID-19 patients.
  4.  Our results suggest that COVID-19 patients discharged from acute care should attend the inpatient PR program, in order to improve their physical function and quality of life including fatigue.'

Please see our changes in the attached manuscript.

Thank you for your suggestions again.

Best regards,

Gilbert Büsching and Zhongxing Zhang
